# The Obesity–Epigenetics–Microbiome Axis: Strategies for Therapeutic Intervention

**DOI:** 10.3390/nu17091564

**Published:** 2025-05-01

**Authors:** Shabnam Nohesara, Hamid Mostafavi Abdolmaleky, Ahmad Pirani, Giuseppe Pettinato, Sam Thiagalingam

**Affiliations:** 1Department of Medicine (Biomedical Genetics), Boston University Chobanian & Avedisian School of Medicine, Boston, MA 02118, USA; samthia@bu.edu; 2Department of Medicine, Division of Gastroenterology, Beth Israel Deaconess Medical Center, Harvard Medical School, Boson, MA 02215, USA; gpettina@bidmc.harvard.edu; 3Mental Health Research Center, Psychosocial Health Research Institute, Iran University of Medical Sciences, Tehran 14535, Iran; a.pirani85@gmail.com; 4Department of Pathology & Laboratory Medicine, Boston University Chobanian & Avedisian School of Medicine, Boston, MA 02118, USA

**Keywords:** obesity, gut microbiome, epigenetics

## Abstract

Obesity (OB) has become a serious health issue owing to its ever-increasing prevalence over the past few decades due to its contribution to severe metabolic and inflammatory disorders such as cardiovascular disease, type 2 diabetes, and cancer. The unbalanced energy metabolism in OB is associated with substantial epigenetic changes mediated by the gut microbiome (GM) structure and composition alterations. Remarkably, experimental evidence also indicates that OB-induced epigenetic modifications in adipocytes can lead to cellular “memory” alterations, predisposing individuals to weight regain after caloric restriction and subsequently inducing inflammatory pathways in the liver. Various environmental factors, especially diet, play key roles in the progression or prevention of OB and OB-related disorders by modulating the GM structure and composition and affecting epigenetic mechanisms. Here, we will first focus on the key role of epigenetic aberrations in the development of OB. Then, we discuss the association between abnormal alterations in the composition of the microbiome and OB and the interplays between the microbiome and the epigenome in the development of OB. Finally, we review promising strategies, including prebiotics, probiotics, a methyl-rich diet, polyphenols, and herbal foods for the prevention and/or treatment of OB via modulating the GM and their metabolites influencing the epigenome.

## 1. Introduction

Obesity (OB) is known as a multifactorial trait characterized by an abnormal accumulation of fat mass and a high body mass index (BMI) [1,2]. One of the main public health problems around the world is the ever-increasing prevalence of OB due to altered lifestyles, regular use of high-calorie foods, and insufficient physical activities, the core causative factors of metabolic disorders [3,4]. Excessive weight also increases the risk for the development of some serious disorders like cancer, nonalcoholic fatty liver disease (NAFLD), type 2 diabetes (T2DM), and cardiovascular diseases [5,6,7]. Additionally, the adipose tissues in obese subjects comprises an elevated amount of classically activated M1 macrophages and reduced amounts of activated M2 macrophages, which produce great amounts of pro-inflammatory cytokines, like tumor necrosis factor alpha (TNF-α), interleukin-1 beta (IL-1β), and interleukin 6 (IL-6), which in turn give rise to the initiation of systemic inflammation and insulin resistance [8,9]. OB-induced inflammation is linked to epigenetic aberrations in adipose tissue. For example, DNA hypermethylation at the peroxisome proliferator-activated receptor γ (PPARγ) 1 promoter induced by OB-dependent factors is a key regulator in the activation of adipose tissue macrophages, leading to pro-inflammatory phenotype, triggering inflammation, and insulin resistance in OB [10]. Furthermore, it has been shown that suppressing M2 macrophage polarization in OB-induced inflammation occurred by adipocyte-derived exosomal microRNA (miRNA)-1224 through Musashi RNA binding protein 2 (MSI2)-mediated Wnt/β-Catenin axis [11]. Remarkably, experimental evidence also indicates that OB-induced epigenetic modifications in adipocytes can be registered in the cellular “memory”, predisposing individuals to weight regain after caloric restriction and subsequently inducing the liver inflammatory pathways [12].

In addition to epigenetic abnormalities, aberrations in the composition and diversity of body microbial communities due to environmental factors and diet can contribute to the development of OB and related metabolic diseases [13]. For example, obese children exhibited an elevated abundance of *Proteobacteria* at the phylum level and decreased abundance of *Oscillibacter* and *Alistipes* at the genus level with corresponding reduction in the concentrations of butyrate and isobutyrate in their fecal samples [14]. Moreover, subjects with T2DM and OB exhibit alterations in the structure of microbial community of the saliva, like reduction in the bacterial richness and diversity that is linked to the consumption of specific foods [15].

The human gastrointestinal (GI) tract is one of the biggest organs in the body, with a total surface area of about 250–400 m^2^, which is the habitat of an intricate and dynamic population of microorganisms involved in the homeostasis or pathogenesis of many illnesses [16,17]. The total number of microorganisms in the human GI tract is about 10^14^ and their collective genomic content is almost 100 times more than the human genome [18]. High-throughput sequencing technologies have demonstrated remarkable reductions in the alpha diversity and changes in the composition at the phylum or genus level of the gut microbiome (GM) in OB [19]. NAFLD, one of the most common consequences of OB is also associated with significant GM and epigenetic alterations [20,21]. In this line, recent studies have elucidated the complex interactions between OB, epigenetic alterations, and GM in NAFLD development [22]. Moreover, gut dysbiosis can enhance intestinal permeability, allowing the translocation of harmful substances to the liver, thus exacerbating liver inflammation and fibrosis [23]. These interconnected mechanisms provide valuable insights into potential therapeutic strategies, including targeting epigenetic modifications, modulating GM, and developing personalized dietary interventions to mitigate OB and the progression of NAFLD and its associated metabolic complications.

In this review, we briefly discuss the potential role of epigenetic aberrations in the onset and development of OB. Then, we review the correlation between abnormal changes in the composition of GM and OB and the interactions between the GM and the epigenome in the development of OB. At the end, we discuss how promising strategies, such as dietary modifications and administration of probiotics, prebiotics, and polyphenols can contribute to the prevention and/or treatment of OB through modulating the microbiota and their epigenetic metabolites. Figure 1 provides an overview of the interconnected topics to be explored in this work.

The gut microbiota metabolizes dietary components, producing bioactive compounds such as folic acid, vitamin B12, and short-chain fatty acids (SCFAs), which influence the epigenome and gene expression. Gut dysbiosis not only disrupts the synthesis of these bioactive compounds but also triggers inflammation, compromising intestinal barrier integrity. This allows harmful metabolites to translocate into the bloodstream, activating immune cells and initiating a cytokine surge. The resulting inflammatory response disrupts metabolic gene regulation in the liver, pancreas, and other tissues, contributing to obesity and metabolic syndrome.

## 2. Epigenetics and Development of Obesity

Epigenetic mechanisms are involved in genes expression fine-tuning without changing the nucleotide sequences [24]. Such mechanisms are capable of regulating the genes activities at the transcriptional and post-transcriptional levels and/or at the translation and post-translational levels, which in turn lead to more diversity in morphogenesis, cell differentiation, stress responses, and adaptability of an organism to the everchanging environmental conditions [25,26,27]. Similarly to genetic codes, epigenetic marks are heritable but flexible and can guarantee the precise transmission of tissue-specific chromatin states and gene expression profiles across numerous cell generations [28]. The main epigenetic mechanisms discussed in this work, include DNA methylation, histone modifications, and RNA interference which is mediated by non-coding RNAs [29]. DNA methylation (DNAmet) involves the transfer of a methyl group onto the C5 position of the cytosine by the mediation of DNMTs (DNA methylal transferase enzymes) to form 5-methylcytosine (5-mC), which further can be modified to 5-hydroxymethylcytosine (5-hmC) by TET enzymes (Ten-Eleven Translocation methylcytosine dioxygeneases) during demethylation reaction. DNAmet contributes to modulation of gene expression by turning genes “on” or “off” in an age- and tissue-related fashion largely through recruitment of DNA binding proteins and suppressing transcription factor binding to the corresponding DNA. As a consequence of a dynamic process involved in both de novo DNAmet and demethylation, methylation patterns are changeable during development, while differentiated cells possess a unique DNAmet pattern capable of modulating tissue-specific gene transcription [30]. Likewise, acetylation/deacetylation or methylation/demethylation of the amino acids of diverse histone proteins tails are versatile post-translational modifications that affect genes expression levels. In this process, the addition or removal of acetyl groups is performed by lysine acetyltransferases and lysine deacetylases, respectively, where acetylation induces but deacetylation suppresses gene expression. Histone methylation can induce or suppress gene expression depend on the amino acid position [31,32].

Non-coding RNAs (ncRNAs) are a heterogeneous class of transcripts, which are not translated into proteins. However, ncRNAs, in particular miRNAs, are involved in a myriad of cellular processes such as RNA maturation and processing, metabolism, signaling, cell proliferation and differentiation, gene expression, and protein synthesis [33,34]. In the following section, we briefly discussed the relationship between epigenetics and the development of OB.

### 2.1. Altered DNAmet in Obesity

OB is connected to derangements in transmethylation and one-carbon metabolism in various body organs. Multiple DNAmet sites play key roles in increasing the risk of OB. Hypomethylation at the cg21178254 site upstream of CCNL1 elevates the risk of OB through escalating the expression of this gene [35]. Hypermethylation at cg02814054 contributes to OB via reducing the MAST3 expression, while hypomethylation at cg06028605 elevates the risk of OB by reducing the SLC5A11 expression [35]. Likewise, rare variants within the 2p23.3 genome site influence OB by increasing susceptibility of the cg01884057 site to methylation, which in turn decreases the expression of POMC, ADCY3, and DNAJC27 [35].

Consumption of an obesogenic diet can contribute to the onset and development of OB and other related disorders by causing abnormalities in DNAmet [36,37]. For example, Vander Velden et al. found that consumption of an obesogenic diet could cause differentially methylated regions (DMRs) in the hippocampus of male mice, which in turn altered the expression of some genes, including BIN1 and histone deacetylase 5 (HDAC5) [38]. Wu et al. reported that consuming a high-fat diet (HFD) by male mice caused disturbance in the gluconeogenesis of offspring by changing Igf2/H19 DNAmet in the liver [39]. HFD-induced OB in mice led to the induction of the expression of DNA methyltransferase-1 (Dnmt1) and Dnmt3a enzymes, elevating DNAmet in the teste, but reducing global DNAmet in the ovaries [40]. Another study by the same group also showed increase in the total expression of Dnmt1, Dnmt3a, and Dnmt3b and a decrease in Dnmt3L and global DNAmet level in the uterus of mice with HFD-induced OB [41]. Additional studies on the association between the consumption of obesogenic diet and abnormalities in DNAmet, and OB in humans are summarized in Table 1.

### 2.2. Histone Modifications in Obesity

In addition to DNAmet, the development of OB is linked to changes in histone modifications. For example, Guo et al. found a total of 281 differentially expressed proteins and 147 differentially modified proteomic and lysine acetylation sites in subjects with metabolically unhealthy OB [57]. Elevated acetylation in visceral adipose tissue of subjects with metabolically unhealthy OB was characterized by specific differentially acetylated sites, such as H3.7K80, H1.2K63, and H1XK90 on histone proteins [57]. As another example, Wu et al. reported that the monocytes of individuals with OB showed elevated levels of mitochondrial acetyl-CoA acetyltransferase 1 and histone acetylation [58]. Alawathugoda et al. recently examined the impact of maternal OB on the genes expression levels and cell fate during embryonic cortical neurogenesis and found that HFD could change EZH2 expression/phosphorylation and reduce H3K27me3 level and associated transcriptional derepression of EZH2 target genes, which in turn disrupted embryonic neurogenesis in rats [59]. More studies on histone modifications in OB are summarized in Table 2.

### 2.3. Altered miRNAs in Obesity

The adipose tissue is one of the core sources of circulating miRNAs, which are involved in adipogenesis, lipid metabolism, and homeostasis [67]. Therefore, the adipose-derived miRNAs levels are considered worthy biomarkers for forecasting the development of OB in adulthood, and for identifying the degree of adipose tissue inflammation [68,69]. In this line, Ahmad Mir et al. identified differences in the expression of a total of 64 miRNAs between subjects with metabolically healthy OB in comparison with metabolically unhealthy OB in the blood samples. In their study, miR-130b-5p, miR-363-3p, and miR-636 were connected to cholesterol level, while miR-130a-3p was linked to low-density lipoprotein (LDL) level [70]. Other examples of altered expression of miRNAs in OB are summarized in Table 3.

While various epigenetic alterations are associated with OB, the epigenome itself is modulated by multiple internal and external factors, like endocrine disruptors, nutritional factors, pharmaceuticals, and inorganic chemicals [77]. The relationship between GM and OB will be addressed after first elucidating the links between GM and OB.

## 3. Gut Microbiota and Obesity

In addition to genetic factors and lifestyle, changes in the GM and oral microbiome are also involved in the development of OB [78]. For example, the prevalence of OB among 647 obese individuals (compared to 969 non-obese subjects) from African American populations was associated with five bacterial taxa in *Firmicutes* and two bacterial taxa in *Actinobacteria and Proteobacteria* [79]. Gut dysbiosis, characterized by an increased abundance of *Firmicutes* and an altered *Firmicutes*/*Bacteroidetes* ratio, has been implicated in metabolic dysfunctions affecting overall systemic homeostasis.

OB can also influence cellular turnover of the intestine by elevating cell death and changing the expression of genes involved in survival/proliferation, which in turn lead to heightened intestinal permeability, alterations in villi/crypt length, and reductions in the expression of genes responsible for mucus synthesis and tight junctions [80]. More investigations into the association between changes in the oral and GM and the development of OB are shown in Table 4.

## 4. Mutual Connections Between Gut Microbial Changes and Epigenetic Alterations

Alterations in the composition of GM involved in the pathogenesis of OB are linked to epigenetic modifications [95]. In this regard, it has been shown that gut bacterial communities by modulating genes expression through epigenetic mechanisms can influence key regulators of glucose and energy homeostasis, such as *HDAC7* and *IGF2BP2* in the adipose tissue [96]. For example, obese subjects with low *Bacteroidetes*-to-*Firmicutes* ratio exhibited DNA hypomethylation and up-regulation of *HDAC7* and *IGF2BP2* genes in the adipose tissue versus individuals with high *Bacteroidetes*-to-*Firmicutes* ratio [96]. As another example, Salas-Perez et al. reported decreased abundance of *Ruminococcus* in obese individuals that could influence BMI via altering DNAmet, particularly in a DMR located at the *MACROD2* gene [97]. Moreover, accumulation of the visceral fat induces the degradation of short-chain fatty acids (SCFAs) and produces lipopolysaccharide (LPS), which in turn increases systemic inflammation [98]. The obesogenic diet is also capable of re-shaping GM, changing bacterial metabolite production, altering histone methylation and acetylation, and subsequently activating pathways involved in the development of colon cancer [99]. Furthermore, a significant difference in colonic acetate production, a known HDAC inhibitor, was observed in lean versus obese youth, which was attributed to variations in colonic lactulose fermentation [100].

Other epigenetic metabolites of the GM are strongly linked to BMI. For example, butyrate was negatively connected to BMI, visceral fat area, and the percentage of body fat in obese individuals [101]. In addition to altering histone methylation and acetylation, SCFAs may exert their beneficial or detrimental effects in OB via changing DNAmet levels. For example, propionate may induce DNA hypermethylation at the cg26345888 locus, leading to the suppression of DAB1 expression in obese individuals. This epigenetic modification has been associated with vitamin D deficiency, which in turn may contribute to the development of diabetes [102].

## 5. Therapeutic Strategies for Prevention or Treatment of Obesity by Microbiome Mediated Epigenetic Modulations

A fundamental therapeutic approach for OB and its associated metabolic disorders is lifestyle modification. In this regard the first and most important remedy is caloric restriction (CR) and physical activity [103], which appears to be difficult for many individuals, in particular in industrialized countries. Besides changing lifestyle, emerging evidence has shown that microbiome-based therapeutics such as the use of probiotics, prebiotics, and some specific diets, may help prevent or manage OB by modulating the oral and fecal microbiome via epigenetic mechanisms. This section will review these findings.

### 5.1. Caloric Restriction (CR) and Physical Activity and Their Influence on Gut Microbiome

CR is the most common option for the prevention or treatment of OB through promoting metabolic health [104,105]. In addition to fat burning, CR can prevent OB by modulating the composition of gut microbiota via increasing probiotic genera like *Bifidobacterium* and *Lactobacillus* and reducing the abundance of *Helicobacter* [106]. Ott et al. reported that a 4-week CR could improve weight loss, enhance gut barrier integrity and alleviate systemic inflammation in women with OB, possibly via epigenetic mechanisms [107]. Chen et al. reported that the anti-OB effects of CR are associated with the establishment of a gut microbiome profile opposing OB. Specifically, they identified an increased abundance of *Christensenellaceae*, a bacterial family involved in acetate and butyrate production. This microbial shift may serve as a potential biomarker for OB reduction [108]. Dong et al. found that high protein CR diet could contribute to weight loss in obese subjects by elevating microbial diversity, increasing the abundance of *Akkermansia* spp. and *Bifidobacterium* spp. and depleting of *Prevotella* spp. [109]. It is important to note that, while CR exhibits beneficial effects in metabolic disorders and OB, excessive CR can lead to liver injury and exacerbate hepatic inflammation by increasing the expression of pro-inflammatory cytokines [110].

Physical activity or exercise also contribute to improving gut dysbiosis in OB and reducing fat accumulation by modulating the composition of GM and their epigenetic metabolites [111]. Exercise has been shown to enhance the abundance of butyrate-producing fecal bacteria and stimulates lipid metabolism through the butyrate-SESN2/CRTC2 signaling pathway [112]. Additionally, Nagano et al. demonstrated that exercise mitigates HFD-induced OB in mice by increasing the relative abundance of butyrate-producing bacterial taxa, such as *Ruminococcaceae* and *Eubacteriaceae*, while reducing the abundance of *Erysipelotrichaceae* and *Rikenellaceae* [113]. In humans, a 12-week combined strength and endurance training program in obese children led to an increased abundance of butyrate-producing gut microbiota, including *Blautia*, *Dialister*, and *Roseburia*, while concurrently down-regulating the OB-associated NLR family pyrin domain containing 3 (NLRP3) signaling pathway [114]. Eight weeks of exercise training in obese individuals could also improve insulin sensitivity and reduce visceral adiposity by increasing the abundance of the genera *Ruminococcus gauvreauii*, *Lachnospiraceae* FCS020 group, and *Anaerostipes* [115]. Furthermore, Qian et al. found that exercise training in obese subjects could reduce the levels of *Akkermansia*, *Intestinimonas* microbiome and elevate the levels of *Ruminococcaceae* UCG011, and *Holdemania* microbiome [116].

Different intensities of exercise may have various impacts on the composition of GM and butyrate-producing bacteria. For example, lower exercise intensity elevated relative abundance of *Bifidobacterium*, *A*. *municiphila*, and butyrate-producers *Lachnospira eligens*, *Enterococcus* spp., and *Clostridium Cluster IV*, while higher exercise intensity escalated the abundance of methane producer *Methanobrevibacter smithii*, and butyrate-producers, including *Eryspelothrichales* and *Oscillospirales* [117].

### 5.2. Dietary Methyl Donors and GM

Dietary methyl donors play a key role in the modulation of glucose and lipid metabolism. Individuals with OB and greater dietary intakes of methyl donors, including betaine, methionine, choline, and vitamins B6, B9, and B12, are less susceptible to being metabolically unhealthy [118]. These ingredients may be considered as good resources to prevent OB via regulation of the GM and epigenetic mechanisms. As an example, Du et al. found that betaine supplementation in mice could improve gut dysbiosis caused by HFD and it is able to increase the abundance of anti-OB strains, including *Bifidobacterium*, *Lactobacillus*, and *Akkermansia muciniphila*. *Akkermansia muciniphila* is one of the master modulators of betaine effects in promoting microbiome ecology and elevating the levels of strains involved in SCFAs production, specially acetate and butyrate [119]. They demonstrated that acetate and butyrate attenuate the development of OB and glucose intolerance by modulating DNA methylation at the promoter of the host miR-378a [119]. As another example [120] demonstrated that a 4-week dietary supplementation with betaine reduced OB in HFD mice by restoring the balance of glucose and lipid metabolisms. This effect was accompanied by the reversal of elevated levels of miR-27a, miR-27b, miR-34a, miR-200b, and miR-223 in the liver of obese mice [120].

### 5.3. GM-Derived Metabolites for Treatment of Obesity

#### 5.3.1. Short Chain Fatty Acids (SCFAs) for Treatment of Obesity

GM-derived metabolites such as butyrate and acetate not only act as inhibitors of HDAC activity but also play key roles in reducing OB-induced inflammation and maintaining gut barrier integrity via regulating the expression of tight junction proteins. These metabolites are also involved in promoting fat thermogenesis, improving mitochondrial function, reducing endoplasmic reticulum stress, hampering autophagy dysfunction, and elevation of the anti-inflammatory M2 macrophages [121,122,123,124].

Additionally, butyrate, as one of the most abundant GM metabolites, plays a key role in decreasing appetite and activation of brown adipose tissue through the gut–brain neural circuit by inhibiting the action of orexigenic neurons involved in the expression of neuropeptide Y in the hypothalamus and reducing the activity of neurons in the nucleus tractus solitarius and dorsal vagal complex in the brainstem [125]. Butyrate and propionate have demonstrated ability for hampering diet-induced OB in mice by stimulating gut hormones and suppressing food intake as well [126]. In this context, sodium butyrate (SB)-treated obese mice exhibited reduction in diet-induced OB, which was associated with increased level of H3K9Ac on the promoter of Adipor1/2, Ucp2 and Ucp3 genes, reduced level of HDAC1, and elevated expression of AMP kinase (AMPK) and adiponectin receptors (adipoR1/2) in muscles [127]. As another example, a two-months butyrate intervention in mid-adult obese mice alleviated neuroinflammation, liver fibrosis, and other adverse impacts of OB [128]. Butyrate also increases glucagon-like peptide-1 (GLP-1) production in colonic cells that is known for its anti-OB effects [129].

In other studies, Mollica et al. found that butyrate could promote mitochondrial cell energy metabolism, respiratory capacity, fatty acid oxidation, and glucose homeostasis in insulin-resistant obese mice by changing the mitochondrial dynamic toward fusion [130]. Furthermore, Wanjun et al. reported that SB administration could inhibit body weight gain, restore the GM composition altered by a HFD, and strengthen intestinal barrier integrity [131]. These effects led to a reduction in serum LPS concentrations (53% lower compared to the HFD group) and a decrease in inflammation [131]. The protective effect of butyrate against OB-induced intestinal barrier disruption is associated with the induction of Paneth cell α-defensin expression, which occurs through histone deacetylation and signal transducer and activator of transcription 3 (STAT3)-dependent mechanism [132]. SB intervention for 12 weeks could also decrease body weight in OB-prone and OB-resistant rats through overexpression of antioxidant enzymes, elevating glutathione (GSH)/glutathione disulphide (GSSG) ratio, reducing reactive oxygen species (ROS) and malondialdehyde (MDA) levels, increasing the expression of Pi3k, Nrf2, Nqo-1, and Ho-1, and reductions in the Gsk-3β mRNA expression [133]. Furthermore, Fu et al. reported that SB supplementation could decrease the body fat content and blood lipids in ovariectomized mice by inducing estrogen receptor alpha (ERα) expression, promoting mitochondrial aerobic respiration, and enhancing the phosphorylated form of AMPK and PGC1α [134].

Butyrate can also suppress quinolinic acid-induced BDNF reduction and OB-induced cognitive dysfunction through epigenetic enhancement of H3K18ac at BDNF promoter region [135]. In another interesting study, Shon et al. found that anti-OB effect of butyrate was associated with reshaping gut microbiome and epigenetic modulation of muscular circadian clock [136]. In their study, butyrate was found to reduce weight gain in HFD rats by increasing the abundance of *Firmicutes* and upregulating the expression of muscle circadian clock genes (Clock, Arntl, and Per2). This effect was linked to increased transcription of genes involved in fatty acid oxidation, which was associated with elevated pan-histone acetylation at the promoter regions of circadian clock genes, likely due to the inhibition of histone deacetylases [136].

Notably, engineered butyrate-producing bacteria may be considered, amongst other promising remedies to hamper OB. For example, engineered *Bacillus subtilis* SCK6 could reduce food intake and body weight gain in HFD mice by improving butyric acid production [137]. The engineered butyrate-producing strain BsS-RS06551 could also hamper OB in mice by improving insulin resistance and glucose tolerance as well as modulating the composition and structure of the GM [138].

In addition to butyrate, human acetate produced by GM and diets containing acetate, like vinegar, have demonstrated the ability to improve insulin sensitivity and glucose homeostasis. From a mechanistic perspective, acetate is capable of promoting the performance of adipose tissue by elevating oxidative capacity and regulating glucose-stimulated insulin secretion in the pancreas [139].

Dietary supplementation of sodium acetate (SA) could reduce appetite and fat deposition in HFD mice by regulating genes and hormones relevant to mitochondrial function, adipogenesis, lipolysis, and beige adipogenesis [140]. In a 12-week intervention in HFD-fed mice, Den Besten et al. reported that oral SA supplementation present in the diet (at 5%, weight/weight proportion) could hamper HFD-induced weight gain (~30%) versus control HFD-fed mice [141]. Likewise, in a study by Lu et al., 16 weeks of oral SA supplementation (5%, wt./wt.) could prevent HFD-induced weight gain by 72% versus control HFD-fed mice [142]. Another study demonstrated that a six-week intragastric administration of acetic acid (50, 250 mmol/L) to HFD-fed mice could contribute to the reduction in weight gain (7% and 8%, respectively) and body fat accumulation than HFD alone [143]. In a human study by Petersen et al., rates of endogenous acetate turnover were ∼30% higher in the lean individuals than obese individuals [144].

SA is capable of decreasing serum triacylglycerol, inflammatory markers like IL-6, free fatty acids, and glucose, as well as elevating serum GLP-1, and leptin levels [140]. Furthermore, Wang et al. showed that dietary acetic acid could attenuate HFD-induced OB in mice by changing taurine conjugated bile acids metabolism [145]. Another study demonstrated that both acetate and propionate, as epigenetic modifying metabolites of the GM, could alert T cell polarization and reduce inflammatory markers, including IL-1β and IL-6 in HFD-fed mice [146]. The protective impact of acetate and propionate against systemic inflammation in OB may be linked to the reduction in TNF-α and IL-6 and changes in free fatty acid receptor (FFAR) and HDAC mRNA expression in monocytes and macrophages from obese individuals as well [147]. Additionally, propionate or a mixture of SCFAs could induce the expression of adiponectin and resistin by altering their promoter DNAmet via reducing the expressions of DNMT1, 3a, 3b and MBD2 and suppressing the binding of DNMT enzymes to adiponectin and resistin promoter regions in the adipose tissue of the HFD-fed mice [148]. Weitkunat et al. reported that propionate could attenuate hepatic gene and protein expression of lipogenic enzymes and subsequently hepatic triglyceride concentration and HFD-induced insulin resistance after 22 weeks in mice [149]. In a study, propionate could reduce interferon-γ and interleukin-17 and blunt histone deacetylase activity in CD4+ T cells and nuclear factor κB activity which further could decrease IL-6 release in primary cells from obese individuals [150]. In 2025, Miyamoto et al. found that elevating propionate production by *Acidipropionibacterium acidipropionici*, a propionate-producing bacterium, could promote metabolism in HFD-fed mice, modulate glucose tolerance, and suppress hepatic inflammation through free fatty acid receptor 3 (FFAR3, also termed GPR41) signaling [151].

Figure 2 depicts a summary of events that through which GM-derived metabolites may affect OB.

#### 5.3.2. Indole and Its Derivatives for Treatment of Obesity and Obesity-Related Disorders

Dietary tryptophan can be metabolized into indole and its derivatives by gut microorganisms. The indole derivatives include indole-3-butyric acid (IBA), indole-3-acetic acid (IAA), indole-3-propionic acid (IPA), indoleacrylic acid (IA), and indole-3-aldehyde (I3A). Indole and its derivatives participate in numerous physiological processes such as maintaining intestinal mucosal homeostasis, improving glucose metabolism, modulation of GI barrier function, and preventing gut dysbiosis and endotoxin leakage [152,153,154,155]. HFD accelerates the release of antimicrobial peptides by Paneth cells, which in turn gives rise to suppressing the *Clostridia* growth in the gut and subsequently reducing the generation of the beneficial metabolite IPA. Therefore, indole and its derivatives may possess potential in the management and treatment of diet-induced OB and fatty liver disease by stimulating GLP-1 secretion and targeting some cellular pathways [156,157]. As an example, Wang et al. found that the GM-derived metabolite IPA could increase leptin sensitivity by targeting STAT3 to prevent diet-induced OB [158]. In another study, Wang et al. reported that elevated I3A from the intestine after sleeve gastrostomy could attenuate the M1/M2 macrophage ratio in the liver and hence improve NAFLD in obese subjects [159].

### 5.4. Probiotics

Probiotics are live microorganisms that possess health benefits at proper concentrations by improving gut barrier integrity to prevent the penetration of bacterial antigens and toxic metabolites to blood circulation [160]. Probiotics also contribute to remodeling of the GM and restoring normal performance in subject with OB and gut dysbiosis by introducing beneficial microbes, reducing fecal abundance of pathogenic bacteria, and attenuating the adipose tissue inflammation through restoring the adipose invariant natural killer T cells [161,162,163,164]. Beneficial impact of probiotics against OB and OB-related disorders is through mitigating epigenetic aberrations. For example, probiotic supplementation during pregnancy can reduce the risk of OB in mothers and their infants by influencing DNAmet status of some genes’ promoter regions relevant to OB and weight gain [165]. In this context, Vähämiko et al. found that probiotic supplementation during pregnancy reduced DNAmet levels of 37 gene promoters and increased DNAmet level of one gene promoter in mothers while decreased DNAmet levels of 68 gene promoters in their infants [165]. Moreover, probiotics have demonstrated the ability to reduce body weight and fat accumulation not only through modulation of the GM but also increasing the concentrations of methyl donors such as folic acid. For instance, the administration of *Bacteroides thetaiotaomicron* to HFD mice for 12 weeks suppressed hyperlipidemia and insulin resistance and decreased body weight and fat accumulation by reducing *Firmicutes*/*Bacteroidetes* ratio and elevating gut–liver folate and unsaturated fatty acid metabolism [166].

In addition to DNAmet alterations, beneficial effects of probiotics against OB are connected to targeting histone modifications due to their key roles in the production of butyrate and acetate, as well known HDAC inhibitors. For example, neuroprotective impact of *clostridium butyricum* in managing cognitive impairment in obese mice is mediated by butyrate production [167]. Choi et al. examined the impact of *clostridium butyricum* on HFD-induced intestinal inflammation and reported its ability for ameliorating inflammatory markers (e.g., TNF-α) and restoring butyric acid in stools in obese males, but not in obese female rats [168]. Vu et al. reported that probiotic treatment could improve OB-associated colitis in mice by enhancing intestinal tight junctions, possibly via epigenetic mechanisms [169].

Cai et al. found that *Lactobacillus plantarum* FRT4, a probiotic present in a type of local yogurt, could attenuate the HFD-induced body weight gain, serum cholesterol, triglyceride, liver and fat weight, and the level of alanine aminotransferase (ALT) in the liver of HFD-induced obese mice by elevating microbial diversity and escalating the abundance of species involved in producing SCFAs, including *Butyicicoccus*, *Butyricimonas*, *Alistipes*, *Bacteroides*, *Parabateroides*, *Intestinimonas*, and *Anaerotruncus* [170]. Additionally, Won et al. examined the anti-OB effect of *Lactobacillus sakei* (*L. sakei*) ADM14 administration in a HFD-induced obese mouse model and found that it was capable of reducing weight gain and blood cholesterol levels by improving the *Firmicutes* to *Bacteroidetes* ratio, elevating the levels of *Bacteroides faecichinchillae* and *Alistipes*, and enhancing butyrate production [171]. Joung et al. reported that treatment of HFD-induced obese mice with *Lactobacillus plantarum* K50 for 12 weeks hampered fat accumulation and low-grade inflammation by promoting the GM composition, attenuating the *Firmicutes*/*Bacteroidetes* ratio, and elevating the concentrations of butyrate and isovalerate [172].

Dietary supplementation with probiotic SF68 has also been shown to reduce intestinal inflammation and improve the integrity and functionality of the intestinal epithelial barrier in obese mice. This effect is mediated by an increase in the expression of tight junction proteins and the intestinal butyrate transporter [173]. Furthermore, *Pediococcus acidilactici* (pA1c^®^) supplementation could decrease body weight, reduce OB-related dyslipidemia, and normalize cholesterol profile in Wistar rats by improving the GM richness, reducing inflammation, and elevating the abundance of butyrate-producing bacteria (*Lachnospiraceae* and *Ruminococcaceae*), *Bacillaceae*, and *Dehalobacteriaceae* families and reducing the abundance of *Aerococcaceae* [174]. Hosomi et al. also found that oral administration of *Blautia wexlerae* to mice could mitigate both HFD–induced OB and diabetes by increasing S-adenosylmethionine, acetylcholine, L-ornithine, succinate, lactate, and acetate via reshaping the composition of the GM [175].

In a double-blind, parallel, randomized, controlled clinical trial involving 32 adult women with OB, probiotic treatment (*Bifidobacterium lactis* UBBLa-70) and symbiotic treatment (*Bifidobacterium lactis* UBBLa-70 combined with fructooligosaccharide) were found to reduce inflammation and have beneficial effects on body weight. These interventions were associated with increased levels of pyruvate and alanine, along with reduced citrate level and decreased ^1H NMR lipid signals [176]. In another human study, a 12 weeks treatment of obese children with probiotics containing *L. rhamnosus* bv-77, *Lactobacillus salivarius* AP-32, and *Bifidobacterium animalis* CP-9 improved OB-related gut dysbiosis by reducing LDL, serum total cholesterol and TNF-α levels and elevating the abundance of *Lactobacillus* spp. and *B. animalis*, species involved in the production of SCFAs [177]. Figure 3 shows a summary of events that mediate probiotics effects on the epigenome.

### 5.5. Engineered Probiotics by Synthetic Biology Approaches for Management of Obesity

Although conventional probiotics have demonstrated efficiency for the treatment of OB and OB-related disorders, they face several limitations, including instability in harsh environments, limited targeting precision, and lack of control over the release of therapeutic agents. Microbes can be reprogrammed via synthetic design to create a variety of therapeutic molecules such as enzymes, cytokines, and signaling molecules from the host and bacterial proteins during pathological condition in the host [178].

Synthetic biology and its related tools like synthetic gene circuits, plasmid-based mechanisms, and CRISPR-Cas systems can improve the functionality, stability, and specificity of engineered probiotics in biomedicine for targeted therapeutic delivery [179,180]. The probiotic survival can be ensured by technologies like encapsulation. The gene circuits facilitate programmable, stimulus-responsive therapeutic release of the engineered probiotics. Therefore, engineered probiotics may be considered promising strategies for managing metabolic and endocrine disorders like OB and diabetes [181,182]. The engineered probiotics are considered non-invasive alternatives to insulin infusion and have demonstrated the ability for precise sensing alterations in glucose concentration and contribution to insulin release [183]. Likewise, the engineered probiotics are capable of managing OB through regulation of appetite-regulating hormones, increasing lipid metabolism, reducing cholesterol levels, elevating satiety, and decreasing food intake [184,185]. Genetically modified probiotics may also contribute to managing OB, diabetes mellitus, and metabolic syndromes by targeting the PI3K/AKT/mTOR signaling pathway [186]. Developing synthetic biological lactic acid bacteria (LAB) probiotics is a promising strategy to manage or treat OB and OB-related disorders, possibly through epigenetic mechanisms, since lactic acid can be employed as substrates in the biochemical synthesis of butyric acid [187,188]. As an example, in a study by Duan et al., the human commensal *Lactobacillus gasseri ATCC 33323 (L)*, *a GLP-1(1-37)*–secreting bacterium was engineered to release GLP-1(1-37), which further contributes to reprogramming intestinal epithelial cells into glucose-responsive insulin-secreting cells [189]. They found that daily oral administration effectively reduced hyperglycemia in a rat model of diabetes.

Additionally, researchers have developed bacteria engineered to produce N-acylphosphatidylethanolamines (NAPEs), bioactive lipids with known anti-obesity properties. These engineered bacteria have shown effectiveness without requiring antibiotic pretreatment, and with administration periods as short as two weeks [190]. This approach has demonstrated success using human NAPE-producing enzymes, supporting the potential for clinical translation. Researchers also identified a strain of Bifidobacterium pseudocatenulatum JJ3 that interacts with an engineered bacterium (BsS-RS06551) through five dipeptides. This consortium showed more pronounced effects on alleviating obesity than either strain administered individually [191]. The mechanism involves perturbation of the vitamin B6 metabolism pathway in the gut and enrichment of beneficial Bifidobacterium longum populations [191]. Modern synthetic probiotics target specific metabolic pathways. For instance, these probiotics can produce SCFAs that interact with gut epithelial cell receptors to increase GLP-1 and PYY levels, enhancing satiety or regulate gene expression by upregulating peroxisome proliferator-activated receptor (PPAR)γ and PPARα while downregulating sterol regulatory element-binding protein-1c (SREBP-1c) and fatty acid synthase expression [192]. Synthetic probiotics may also target the gut–liver axis to preferentially regulate visceral and subcutaneous fat deposits [193]. Furthermore, droplet-based microfluidic approaches have enabled the isolation of bacteria that can utilize metabolites from engineered bacteria, facilitating the construction of synthetic microbial consortia with enhanced anti-obesity potential [191]. Considering the novelty of these approaches in the field, and despite their promise, further studies are warranted to assess their long-term and potential evolutionary impacts.

### 5.6. Prebiotics/Postbiotics

Prebiotics are ingredients in diets that boost the proliferation or function of the gut microorganisms [194]. Specific prebiotics are capable of managing OB by promoting intestinal microbial structure and function [195]. In this line, Wei et al. have shown that a dietary polysaccharide from *Enteromorpha clathrata*, a marine-derived green alga, could reduce the body weight and the serum concentrations of triacylglycerol and cholesterol in HFD-fed mice through increasing the abundance of butyrate-producing bacterium, *Eubacterium xylanophilum* in the gut, reshaping the GM structure, and improving the gut dysbiosis [196]. Raspberry polysaccharides, food-derived compounds, have shown abilities for reducing OB and related conditions such as hyperlipemia, hyperglycemia, endotoxemia, inflammation, and oxidative stress in the liver of obese mice by elevating the abundance of *Dubosiella* and its metabolite (butyrate) and enhancing intestinal barrier integrity [197]. Formononetin, an isoflavone, has been shown to normalize body weight and hyperglycemia while improving the high-density lipoprotein (HDL)-to-LDL ratio. This effect is attributed to the increased abundance of specific gut bacteria, including *Clostridium aldenense*, *Clostridiaceae* (unclassified), and *Eubacterium plexicaum*, as well as acetate and butyrate-producing species. Additionally, formononetin modulates liver miRNA expression associated with OB and reduces the levels of inflammatory markers, such as IL-6, IL-22, and TNF-α [198]. Using certain bioactive dietary fibers, such as apple pectin, inulin, β-glucan, xanthan gum, guar gum, xylan, and carrageenan is also a promising strategy to alleviate OB through the enrichment of commensal bacteria, including *Ruminococcus*, *Butyricimonas*, *Akkermansia*, *Oscillospira*, and *Prevotella* and possibly their epigenetic metabolites [199]. As another example, Zheng et al. found that *Dendrobium officinale (D. officinale)* dietary fiber could diminish OB, liver steatosis, inflammatory responses, and oxidative stress in HFD-induced obese mice by reducing the level of *Bilophila*, elevating the levels of *Bifidobacterium*, *Akkermansia*, and *Muribaculum* as well as enriching the concentrations of acetate and taurine [200].

Administration of soybean insoluble dietary fiber in HFD mice for 20 weeks could also normalize body weight, total cholesterol, triglyceride, and diminish hepatic lipid content by heightening the levels of commensal bacteria, including *Lactobacillales*, *Lactobacillus* [genus], *Lachnospirace*_Nk4A136_group [genus]), and decreasing levels of pathogenic bacteria, including *Lachnospiraceae* [family] and *Bacteroides_acidifaciens* [species] as well as increasing SCFAs concentrations [201]. Functional fiber-rich diets also strongly affect GM composition in various intestinal segments of obese mice. For example, Xu et al. reported that functional fiber supplementation could contribute to the transfer and colonization of microbes from the anterior to the posterior intestinal segments and elevate the abundance of bacteria critically involved in the production of acetate and butyrate [202]. Moreover, insoluble dietary fibers derived from brown seaweed *Laminaria japonica* can reduce serum cholesterol and glucose concentrations in obese mice by increasing the levels of *Akkermansia muciniphila* and restoring the cecal propionate and acetate concentrations [203]. Inulin, a natural soluble fiber, is also capable of improving imbalanced glucose and lipid metabolism in obese mice by enriching SCFAs-producing *Ruminococcaceae* and *Lachnospiraceae* bacteria at the family level and elevation of the fecal SCFAs concentrations [204].

Additionally, anti-OB capabilities of Tibetan highland barley fiber in obese mice was attributed to heightening the abundance of *Lachnospiraceae*_NK4A136_group, *Akkermansiaceae*, and *Muribaculaceae*, decreasing the abundance of *Alloprevotella*, *Prevotellaceae*, *Rikenellaceae*, and *Bacteroidaceae*, and increasing the concentrations of acetate, propionate and butyrate [205].

In humans, a randomized controlled trial among overweight or adults with OB showed that avocado (containing dietary fiber and monounsaturated fatty acids) consumption increased bacterial alfa diversity, the abundance of *Faecalibacterium*, *Lachnospira*, and *Alistipes* about 26–65% and increase the fecal concentrations of acetate, stearic acid, and palmitic acid versus controls [206]. Furthermore, Iversen et al. demonstrated that a dietary intervention involving high-fiber rye products could increase the abundance of the butyrate-producing bacterium *Agathobacter*, while decreasing the abundance of the *Ruminococcus torques* group in humans, a shift associated with weight loss. However, the anti-OB potential of corn peptides in combination with swimming has been linked to the elevation of the abundance of beneficial bacteria like *Oscillospiraceae*, reductions in the levels of *Alloprevotella*, and increasing folate synthesis, a methyl donor in one carbon metabolism [207].

Polyphenols are among other nutritional compounds that may function as prebiotics. Among them, resveratrol butyrate esters, derivatives of resveratrol and butyric acid, have demonstrated capabilities for inhibiting bisphenol A (BPA)-induced OB in rats female offspring by decreasing the *Firmicutes*/*Bacteroidetes* ratio and the fecal concentration of acetate [208]. Qiao et al. investigated the protective impact of apigenin (Api) against OB-associated metabolic syndrome and found that Api supplementation could ameliorate intestinal dysbiosis induced by HFD via reducing gut barrier damage, increasing the abundance of *Akkermansia* and *Incertae_Sedis*, involved in acetate and butyrate production, and decreasing the abundance of *Faecalibaculum* and *Dubosiella* at the genus level [209]. Li et al. also found that the anti-OB features of a chlorophyll-rich spinach extract supplementation for 13 weeks in HFD mice was due to elevating levels of *Akkermansia*, a bacterium involved in acetate production [210]. Furthermore, the preventative impact of alkaloids and polysaccharides present in black tea on HFD-induced OB in mice have been associated with improvements in gut dysbiosis and modulation of DNAmet in imprinted genes such as Magel2, Ctnna3, Gab1, and L3mbtl1 in the spermatozoa of the animals [211].

Postbiotics, which are bioactive compounds produced when gut bacteria break down prebiotics, also have significant anti-OB effects. For example, the postbiotics generated through the bioconversion of citrus pomace and whey by kefir LAB, such as *Lactobacillus* spp., may help reduce hypertriglyceridemia and body weight gain in HFD mice. This effect is likely mediated by an increase in the relative abundance of *Akkermansia odorimutans*, which is involved in butyrate production, and a decrease in the abundance of *Odoribacter profusus*, a bacterial marker of OB [212]. Nowadays, postbiotics are increasingly recognized as promising alternatives to probiotics and prebiotics in the realm of gut or body health and microbiome modulation. While postbiotics show great promise, they are not intended to completely replace probiotics and prebiotics. Rather, they offer an additional tool in the growing field of microbiome modulation and gut health optimization. The ideal approach may involve a combination of these three elements, tailored to individual needs and health goals.

### 5.7. Antibiotics and Gut Microbiota Changes

Although antibiotics are known for their disturbing impacts on GM composition, in some conditions, antibiotics have the capacity for changing GM and influencing the expression of some proteins in adipose tissue in HFD mice by altering genes’ promoter DNAmet, which in turn could elevate fatty acid oxidation and reduce body weight gain [213]. For example, Yao et al. found that antibiotic use suppressed the body weight gain after 16 weeks of HFD feeding in mice by altering the composition of GM, improving the beta oxidation and thermogenesis (associated with the elevations in the expression of the related genes, including PPAR-α, Pgc-1α, and Atgl), decreasing DNAmet of the adiponectin and resistin promoters, and reducing the expression of DNMT1 and DNMT3a [213]. Nevertheless, this line of research requires further investigation to draw definitive conclusions.

## 6. Existing Challenges and Research Directions

Although mounting evidence has shown that probiotics and prebiotics are capable of preventing or managing OB, the lack of knowledge of the long-term impact of these microbiome-based therapeutics may hamper their clinical applications. Therefore, future investigations should focus on developing effective long-term preventive strategies with the capacity for inclusion into one’s lifestyle. Additionally, most of the current findings related to the effects of microbiome-based therapeutics in preventing and managing OB have been obtained using animal models, especially rodents. More investigations in humans are needed before microbiome-based therapeutics can be extensively prescribed for the prevention or treatment of OB. Although numerous studies have shown that GM-derived metabolites such as butyrate are capable of preventing OB via epigenetic regulation, some current studies have demonstrated the potential obesogenic role of a large amount of butyrate in early life [214]. For example, as maternal smoking during pregnancy, which increases the risk of OB and overweight in offspring, is connected to an elevated abundance of *Firmicutes* and, subsequently, increased microbial butyrate production [215]. Therefore, numerous factors such as age and other environmental factors should be considered to establish successful GM-targeted therapeutic approaches for the prevention or management of OB. Another major challenge is that findings on the association between the microbiome and OB development remain relatively weak, often confounded by small sample sizes and a high degree of inter-individual variability. Therefore, large-scale investigations with larger sample sizes are essential to precisely clarify the host metabolic-bacterial cross-talk and the interplays between GM and host genetics in the development of OB and OB-related disorders.

Microbiome research also faces several methodological limitations that can impact the interpretation of results. These include sequencing approach differences, standardization issues, and challenges in determining causation. 16S rRNA sequencing involves amplifying and sequencing the 16S rRNA gene, a marker specific to bacteria and archaea. This cost-effective method provides a broad overview of microbial community composition. However, it offers limited taxonomic resolution (often only to the genus level) and does not provide functional information. It can also suffer from PCR bias, where certain sequences are preferentially amplified. Metagenomic sequencing (whole-genome shotgun sequencing) sequences the entire DNA content of a sample, providing species-level resolution and revealing the functional potential of the microbial community by identifying genes and metabolic pathways. However, metagenomic sequencing is more expensive and requires greater computational resources. It can also be biased toward organisms with larger genomes or those that are more easily lysed [216]. Therefore, the choice of sequencing method affects the level of detail and type of information obtained. 16S rRNA sequencing is suitable for broad community surveys, while metagenomics is necessary for in-depth functional analysis. Critically, insufficient sequencing depth can lead to underestimation of microbial diversity and inaccurate relative abundance estimates, regardless of the method used.

In both methods, variability can be introduced by factors such as sampling site, time of day, and collection method. For example, stool samples may not fully represent the gut microbiome, and swabs can be affected by environmental contaminants. Standardizing these factors across studies is challenging due to logistical and practical constraints. Furthermore, DNA extraction methods can selectively recover DNA from certain microbial groups, leading to biased results, and improper storage (e.g., temperature fluctuations and long storage times) can alter microbial composition and DNA integrity [217]. The lack of standardization can lead to inconsistencies between studies, making it difficult to compare results and draw general conclusions. Furthermore, most microbiome studies are observational, which complicates determining whether changes in the microbiome cause a particular outcome or are simply correlated with it. Many factors (e.g., diet, genetics, and environment) can also influence both the microbiome and the outcome of interest, leading to spurious associations.

In respect to intervention studies, while various interventions (e.g., fecal microbiota transplantation and dietary interventions) can provide stronger evidence for causation, they are often complex and may have unintended effects. Even when a causal link is established, the underlying mechanisms may remain unclear. Therefore, without careful experimental design and mechanistic validation, determining whether the microbiome plays a causal role in a particular disease or condition remains difficult. Eventually, these methodological limitations can compromise the validity and reproducibility of microbiome research.

In summary, due to the necessity for diverse resources and interdisciplinary teams and their high costs, exploring the microbiome’s association with OB development can be challenging. Nevertheless, basic, translational, and human mechanistic investigations that particularly discover functional associations between the GM and OB and encompass cross-cutting interdisciplinary teams could help drive the field forward. The design of sophisticated bioinformatics tools and innovative and promising technologies capable of integrating the information from both host and microbiome would enhance a systems-level understanding.

## 7. Conclusions

The GM has a dynamic metabolic potential by affecting various metabolic pathways in the host. Recent findings have shown that the pathogenesis of OB is associated with alterations in the microbiota composition and subsequently epigenetic aberrations. The clarification of microbial features relevant to OB across various populations and their interplays with diet, epigenetics, and host genetics paves the way to explore promising strategies with more efficiency and minimal side effects for managing OB and OB-related disorders. Generally, OB and OB-related disorders are associated with low diversity in microbial composition and gene count.

Reshaping the composition of the GM and normalizing epigenetic abnormalities using microbiome-based therapeutics may be considered a promising strategy to obtain stable weight reduction. Moreover, targeting the epigenetic metabolites produced by these microorganisms and the host using dietary intervention and drugs may be considered a promising approach for managing, and preventing OB and OB-related disorders. Promising strategies for the prevention and/or treatment of OB via modifying the GM composition via epigenetic shifts are consuming probiotics, prebiotics, post-biotics, and some specific diets, such as dietary methyl donors and polyphenols. These interventions may be considered the most efficient approaches with minimal side effects to mitigate OB and OB-related disorders in the clinic, as they can selectively promote the number or the growth and activity of health-promoting bacteria and improve the intestinal microecological environment of intestinal bacteria under obese conditions.

However, more detailed investigations are needed to explore the precise mechanisms of action of epigenetic metabolites/alterations and the roles played by the microbial communities in the development and prevention of OB.

## Figures and Tables

**Figure 1 nutrients-17-01564-f001:**
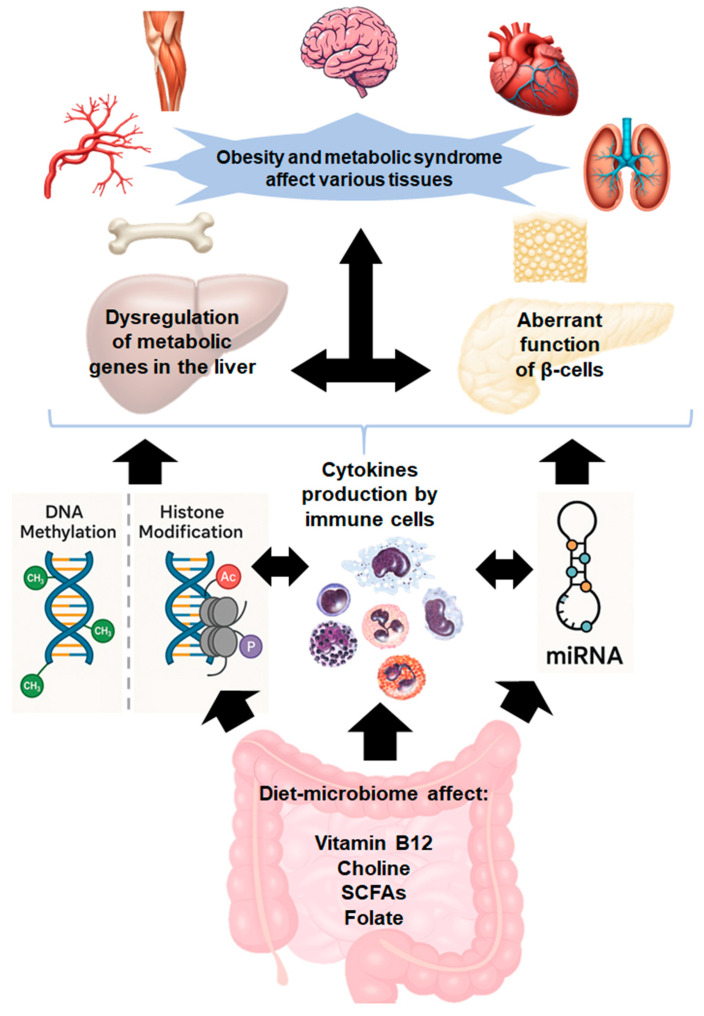
Mechanistic cascade linking gut dysbiosis to obesity and metabolic syndrome.

**Figure 2 nutrients-17-01564-f002:**
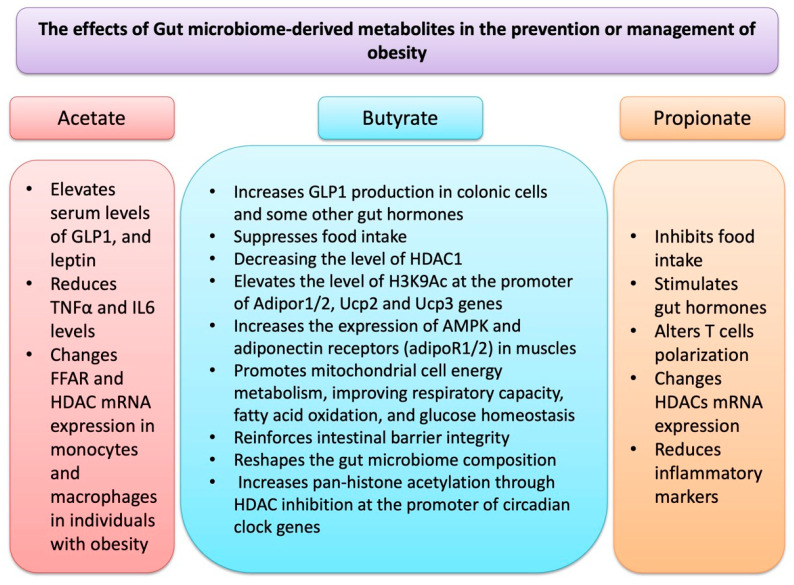
GM-derived metabolites, in particular SCFAs such as acetate, butyrate and propionate affect epigenetic regulation and other molecular events related to obesity.

**Figure 3 nutrients-17-01564-f003:**
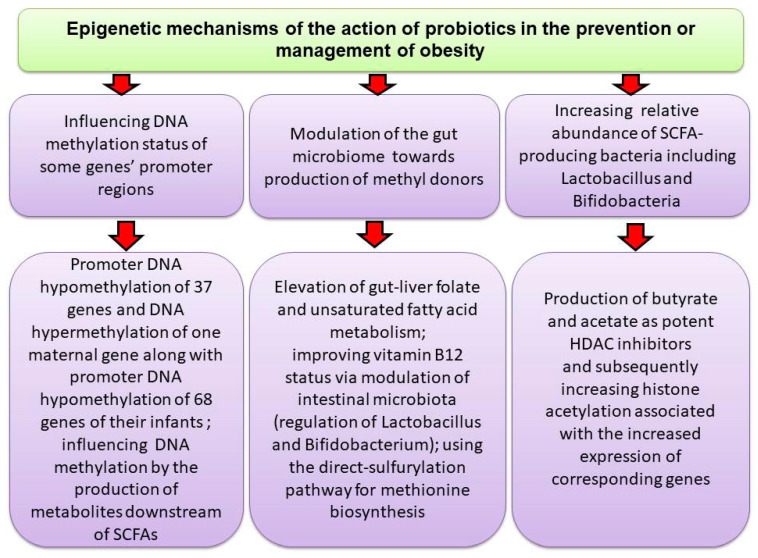
Probiotics through different mechanisms may affect DNAmet or produce SCFAs that promote gene expression by inducing histone acetylation.

**Table 1 nutrients-17-01564-t001:** DNA methylation alterations associated with obesity (OB).

Sample Type	Affected Genes	Main Outcome	Ref.
Sperm	*TP53AIP1*, *SPATA21*, *PTPRN2*, and *ZNF33A*	3264 differentially methylated CpG sites between normal weight and obese men; hypomethylation of *TP53AIP1* and *SPATA21* in obese vs. normal weight men	[42]
Visceral adipose tissue (VAT)	*S100A8* and *S100A9*	Hypomethylation and overexpression of *S100A8* and *S100A9* in obese subjects	[43]
VAT	Angiotensin-converting enzyme 2 (*ACE2*) gene	Hypermethylation of *ACE2* gene in obese vs. normal subjects	[44]
Placenta	*METTL-3*/*-14*, *WTAP*, *RBM15B*, and *KIAA1429*	Elevated levels of 5-methylcytosine (5mC) and reduced activity of Ten-Eleven Translocation (TETs) enzymes; reduced N6-methyladenosine (m6A) levels and *RBM15B*, *WTAP*, and *KIAA1429* expression in the placenta with maternal OB	[45]
Blood	*PTPRN2* and *MAD1L1*	1725 differentially methylated regions (DMRs) in male neonates from women with OB vs. normal weight women (1173 regions hypermethylated and 552 hypomethylated)	[46]
VAT	*ATP11A*, *LPL*, and *EHD2*	11120 differentially methylated CpGs and 96 DMRs in women with OB and T2DM vs. without T2DM	[47]
Liver	*CACNA1B*, *CNR1*, *GNAI3*, *PRKCA*, *GNGT2*, *GNG12*, *ADCY6*, and *DRD2*	Altered DNAmet of 3169 CpGs in OB	[48]
Blood	*PCSK7*, *RNF214*, *SYN3*, *JARID2*, *OCA2*, and *POLR2C*	Lower DNAmet at cg03158092 and cg05985988 sites are linked to insulin resistance and beta-cell function in early pregnancy; DNAmet of cg12082129 and cg11955198 sites correlate with higher insulin concentrations in late pregnancy	[49]
Peripheral blood leukocytes	*GLP1R*	Association between OB and DNAmet of the intronic region of *GLP1R*	[50]
Blood	*TFAM* and *PIEZO1*	Lower levels of DNAmet in obese vs. normal weight children; altered DNAmet of cg05831083 and cg14926485 sites in obese vs. normal weight children	[51]
Blood	*BMAL1*	DNAmet of *BMAL*1 is associated with obese phenotype	[52]
Leukocytes	*AHDC1*, *ANXA7*, *MED12L*, *TBXAS1*, and *ENGASE*	Lower levels of DNAmet in the top differentially methylated positions in OB	[53]
Blood	*TNF-α*	Lower DNAmet of *TNF-α* in OB vs. normal weight individuals	[54]
Cord blood and placenta	*PLIN4*, *UBE2F*, and *PPP1R16B*	Association between DNAmet profiles of certain genes, including *PLIN4*, *UBE2F*, and *PPP1R16B* in cord blood and infant weight	[55]
Leukocytes	*CPT1B*	Higher DNAmet of *CPT1B* gene (involved in lipid oxidation), is linked to lower serum selenium in obese vs. normal weight individuals	[56]

**Table 2 nutrients-17-01564-t002:** Histone modifications alterations related to obesity (OB).

Type of Histone Modification/Study Subjects	Sample Type	Affected Genes	Key Findings	Ref.
Histone acetylation/mouse	Hippocampus	oxytocin receptor (*Oxtr*)	increased H3K9Ac (an active histone mark) binding at the *Oxtr* promoter in male offspring of maternal HFD	[60]
Histone deacetylase enzymes/Human	Visceral adipose tissue (VAT) and subcutaneous adipose tissue (SAT)	*HDAC3* and *HDAC9*	Reduced levels of *HDAC3* in the SAT and *HDAC9* in the VAT in obese vs. normal wight women	[61]
Various histone modifications (acetylation, propionylation, and crotonylation/mouse	Testis	*AKAP4*, *ODF2*, *PRKACB*, *SPAG6*, *LDHC*, *PGK2* and *GAPDHS*	Reduced levels of testicular H4K8ac, H3K122ac, H3K23pr and H4K8cr in obese mice vs. controls	[62]
Histone deacetylation enzyme/mouse	Adipose tissue	*Leptin*	Increased activity of the cytosolic histone deacetylase 6 (HDAC6) in obese mice	[63]
Histone acetylation and methylation/mouse and human	White adipose tissue (WAT)	*Bmal1*, *PPAR-γ*, and *Slc1a5*	Reduced H3K27ac and H3K4me3 at *Bmal1* promoter due to decreased methionine and glutamine levels in obese WAT	[64]
Histone acetylation/Human	Peripheral blood mononuclear cells	*SIRT1*	Higher histone acetylation and decreased expression of *SIRT1* in OB vs. control subjects	[65]
Histone acetylation/mouse	Testis	*GRTH*/*DDX25*, *CRM1*, *HMGB2*, *PGK2*, and *tACE*	Decreased H3AcK18 and H4tetraAck (histone H4AcK5, K8, K12 and K16), and aberrant protamine 1 deposition	[66]

**Table 3 nutrients-17-01564-t003:** miRNA alterations related to obesity (OB).

Affected miRNA/Study Subjects	Sample Type	Affected Genes	Key Findings	Ref.
miRNA Let-7/mouse	Liver	*AMPK*	Overexpression of Let-7 in the newborn mice from obese dams	[71]
miRNA 192/human	Serum	*TNFα*, *IL-1Ra*, and procalcitonin	miRNA 192 upregulation in metabolically unhealthy OB	[72]
miR-582-3p and miR-582-5p/mouse	Liver	*AMPKα*, *SAPK*/*JNK*, *Tgfβ1*, *Map3k14*, *Bax*/*Bcl-2*, and *Col1a1*	Correlation between maternal OB and elevated hepatic miR-582-3p and miR-582-5p	[73]
miR-5099, miR-551b-3p, miR-223-3p, miR-146a-3p and miR-21a-3p/mouse	Kidney	Adiposity-related pro-inflammatory and pro-fibrotic genes (*MCP1*, *RANTES*, *TNFα*, and *iNOS*	Differences in the expression of nine miRNAs upon HFD feeding vs. standard diet	[74]
miR-33b/human	Serum	*ABCA1*, *CROT*, *HADHB*, and *NPC1*	Hyperexpression of miR-33b in obese vs. control subjects	[75]
microRNA-450a-5p/mouse	Serum, liver, and white adipose tissue	* DUSP10 *	Increased expression of microRNA-450a-5p in obese mice	[76]

**Table 4 nutrients-17-01564-t004:** Microbiome alterations associated with obesity (OB).

Subjects/Case # or Condition	Type of Microbiome/Sample	Key Finding	Ref.
Human/lean and obese, young, Chinese	Gut microbiome (GM)/fecal samples	Reduced level of *Bacteroides thetaiotaomicron*, a glutamate-fermenting commensal, in obese vs. normal subjects	[81]
Human/33 adults with OB and 29 normal weight controls	Oral microbiome/saliva	Reduced bacterial diversity and richness in OB; increased abundance of *Solobacterium*, *Mogibacterium*, *Prevotella*, *Granulicatella*, *Peptostreptococcus*, and *Catonella*, and reduced abundance of *Capnocytophaga*, *Haemophilus*, *Corynebacterium*, and *Staphylococcus* in OB	[82]
Human/obese hyperglycemic individuals in Qatar	Oral microbiome/saliva	Increased *Firmicutes*/*Bacteroidetes* ratio and reduced *Fusobacteria* phylum in OB vs. controls subjects	[83]
Mice/HFD-fed mice at 12 weeks	GM/fecal and cecal samples	Reduced abundance of *Lactobacillaceae*, *Bifidobacteriaceae*, *Erysipelotrichaceae* and *Verrucomicrobiaceae* following HFD consumption	[84]
Human/Mexican children, 9–11 years-old (10 normal and 10 obese)	GM/fecal samples	Higher *Ruminococcus* spp. in normal weight but *Prevotella* spp. in OB; 19-fold increase in Human herpesvirus 4 in feces of obese children; inverse relationship between *Oscillospiraceae* family and cholesterol level in OB	[85]
Human/26 subjects (13 normoweight vs. 13 obese)	GM/fecal samples	Increased *Collinsella*, *Clostridium* XIVa, and *Catenibacterium*; decreased *Clostridium sensu stricto*, *Romboutsia*, *Oscillibacter* and *Alistipes* in OB	[86]
Human/Indonesian adults (n = 21)	GM/fecal samples	Reduced bacterial diversity and higher primary bile acids concentration in OB	[87]
Human/21 adults with OB vs. 21 controls	GM/fecal samples	Decreased gut microbiota diversity and *Firmicutes*/*Bacteroidetes* ratio in OB; increased *Megamonas*, *Prevotella*, *Fusobacterium*, and *Blautia* but decreased *incertae_sedis*, *Lachnospiracea_ Gemmiger*, *Clostridium* XlVa, and *Faecalibacterium* in OB	[88]
Human/normo-weight vs. obese	Oral microbiome/saliva	Greater abundance of the *Capnocytophaga* genus in OB	[89]
Human/male and female adults	GM/fecal samples	Elevated *Prevotella*/*Bacteroides* ratio and reduced fecal tryptophan level in OB	[90]
Obese cats and normal weight cats	GM/fecal samples	Reduced diversity and abundance of *Firmicutes*, and reduced ratio of *Firmicutes*/*Bacteroidetes* in obese cats	[91]
Human/Infants of women with OB	GM/stool samples at 1, 6, and 12 months	Reduced levels of SCFA-producing bacteria (*Ruminococcus* and *Turicibacter*) and fecal butyric acid in obese vs. normal infants at 1 month age; decreased levels of *Lachnospiraceae* at 6 months age	[92]
Human/30 obese and 30 normal weight children aged 3–5 years	Oral and GM/saliva and fecal samples	Increased abundance of *Filifactor* and *Butyrivibrio* in the saliva and *Faecalibacterium*, *Tyzzerella*, and *Klebsiella* in the fecal samples in OB	[93]
Human/infants born to obese and normoweight mothers (23/group)	GM/stool samples	Higher *Bacillota*/*Bacteroidota* ratio at 6 months of age with maternal OB	[94]

## Data Availability

Not applicable.

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
