# Peer review of "The Obesity–Epigenetics–Microbiome Axis: Strategies for Therapeutic Intervention"

_nutrients, 2025, doi:10.3390/nu17091564_

Round 1
Reviewer 1 Report
Comments and Suggestions for Authors
The manuscript entitled "The Obesity-Epigenetics-Microbiome Axis: Strategies for Therapeutic Intervention" provides an in-depth review of the multifaceted relationships between obesity, epigenetic modifications, and alterations in the gut microbiome. The topic is highly relevant and timely, particularly given the increasing global prevalence of obesity and the growing interest in personalized and systems-level approaches to its prevention and treatment. The review is generally well-organized and covers a broad range of molecular mechanisms, including DNA methylation, histone modifications, and non-coding RNAs, as well as microbiome dysbiosis and its implications for metabolic diseases such as NAFLD. The inclusion of current literature and well-structured tables is appreciated and enhances the manuscript’s value as a reference.
However, several areas require improvement to meet the standards for publication
- Language and Grammar:
The manuscript contains frequent grammatical errors and awkward phrasings that hinder readability. A thorough professional language edit is strongly recommended to enhance clarity and flow. The manuscript contains numerous grammatical errors, awkward phrasing, and formatting inconsistencies that need careful editing. - Redundancy and Repetition:
Several sentences repeat similar concepts (e.g., the relationship between OB and inflammation, or between OB and epigenetics) without adding new insights. Streamlining these sections would improve conciseness. - Structural Organization: The transitions between paragraphs are sometimes abrupt. Adding subheadings within sections (e.g., “DNA Methylation in Obesity”, “Histone Modifications”, “miRNAs and Adipose Tissue”) would greatly enhance readability.
- The table presentations (e.g., Table 1) are data-rich, but some of the descriptions are unclear or incomplete. Consider summarizing the key findings more clearly in the text to help guide the reader.
- Scientific Depth and Contextualization:
While the manuscript cites many recent studies, the discussion often reads as a list of findings without synthesis. The authors should aim to connect findings across studies to reveal overarching mechanisms or patterns, and discuss inconsistencies in the literature. - References and Formatting Issues: Minor issues with reference formatting and citation consistency are noted (e.g., spacing between citations, placement of reference numbers). While the manuscript includes a large volume of information, it would benefit from deeper critical analysis and synthesis of the cited literature, particularly regarding mechanistic pathways and therapeutic implications.
Author Response
Dear Ms. Lvy Hua
Nutrients Section Managing Editor,
Thank you for providing the reviews of our manuscript and sharing the detailed reports. We greatly appreciate the valuable feedback from the reviewers. We have carefully considered all comments, revised the manuscript accordingly, and addressed each point raised. In the revised version, amendments made in response to the first reviewer's comments are highlighted in green and amendments made in response to the second reviewer's comments are highlighted in blue.
We believe these revisions have significantly improved the quality and clarity of our manuscript. We hope that our revised version will now be considered suitable for publication in your esteemed journal. Please find below a point-by-point response to the reviewers' comments.
Best regards,
Dr. Shabnam Nohesara
Reviewer 1
Comments and Suggestions for Authors
The manuscript entitled "The Obesity-Epigenetics-Microbiome Axis: Strategies for Therapeutic Intervention" provides an in-depth review of the multifaceted relationships between obesity, epigenetic modifications, and alterations in the gut microbiome. The topic is highly relevant and timely, particularly given the increasing global prevalence of obesity and the growing interest in personalized and systems-level approaches to its prevention and treatment. The review is generally well-organized and covers a broad range of molecular mechanisms, including DNA methylation, histone modifications, and non-coding RNAs, as well as microbiome dysbiosis and its implications for metabolic diseases such as NAFLD. The inclusion of current literature and well-structured tables is appreciated and enhances the manuscript’s value as a reference.
We sincerely appreciate your valuable comments. We have carefully revised the manuscript in accordance with your suggestions. Appropriate amendments have been made, and additional information has been included, all of which are highlighted in green for your convenience.
Comment: However, several areas require improvement to meet the standards for publication
- Language and Grammar:
The manuscript contains frequent grammatical errors and awkward phrasings that hinder readability. A thorough professional language edit is strongly recommended to enhance clarity and flow. The manuscript contains numerous grammatical errors, awkward phrasing, and formatting inconsistencies that need careful editing.
Author response and action taken: Based on your comment, the English language of the whole manuscript has been meticulously revised (grammatically, punctuation, spelling mistakes and rephrasing of the confusing sentences) to improve the quality of this manuscript. We performed a throughout checking and implemented the comments in the revised version.
- Redundancy and Repetition:
Several sentences repeat similar concepts (e.g., the relationship between OB and inflammation, or between OB and epigenetics) without adding new insights. Streamlining these sections would improve conciseness.
Author response and action taken: In the revised version we removed repeated concepts. A throughout checking was performed for all sections and Tables.
- Structural Organization: The transitions between paragraphs are sometimes abrupt. Adding subheadings within sections (e.g., “DNA Methylation in Obesity”, “Histone Modifications”, “miRNAs and Adipose Tissue”) would greatly enhance readability.
Author response and action taken: We implemented the comment in the revised version and added subheadings within section 2 and section 5 as explained below.
- Epigenetics and development of obesity
2.1 Altered DNAmet in Obesity
2.2 Histone Modifications in Obesity
2.3 Altered miRNAs in Obesity
- Therapeutic strategies for prevention or treatment of obesity by mi-crobiome mediated epigenetic modulations
5.1. Caloric restriction (CR) and physical activity and their influence on gut microbiome
5.2. Dietary methyl donors and GM
5.3. GM-derived metabolites for treatment of obesity
5.3. 1. Short chain fatty acids (SCFAs) for treatment of obesity
5.3. 2 Indole and its derivatives for treatment of obesity and obesity-related disorders
5.4. Probiotics
5.5. Engineered Probiotics by Synthetic Biology Approaches for Management of Obesity
5.6. Prebiotics/ Postbiotics
5.7. Antibiotics and gut microbiota changes
- The table presentations(e.g., Table 1) are data-rich, but some of the descriptions are unclear or incomplete. Consider summarizing the key findings more clearly in the text to help guide the reader.
Author response and action taken: In the revised version we summarized the text and provided the key findings as suggested. Please see Table1, 2, and 3
- Scientific Depth and Contextualization:
While the manuscript cites many recent studies, the discussion often reads as a list of findings without synthesis. The authors should aim to connect findings across studies to reveal overarching mechanisms or patterns, and discuss inconsistencies in the literature.
Author response and action taken: We revised the discussion section to better connect findings across studies, aiming to highlight overarching mechanisms or patterns. Additionally, we added several sentences at the end of relevant sections, as indicated.
References and Formatting Issues: Minor issues with reference formatting and citation consistency are noted (e.g., spacing between citations, placement of reference numbers). While the manuscript includes a large volume of information, it would benefit from deeper critical analysis and synthesis of the cited literature, particularly regarding mechanistic pathways and therapeutic implications.
Author response and action taken: We revised the formatting and ensured citation consistency based on your helpful comment. We also conducted a thorough review and added more information on mechanistic pathways and therapeutic implications, all highlighted in green.
Reviewer 2 Report
Comments and Suggestions for Authors
This study systematically reviewed the interactions between obesity, epigenetics and gut microbiome (GM), covering multidimensional mechanisms such as DNA methylation, histone modification and miRNA regulation. Detailed list of microbiome changes related to obesity (such as Firmicutes/Bacteroidetes imbalance) and treatment strategies (such as probiotics, SCFAs, calorie restriction, exercise). At the same time, the tables are organized clearly (such as Table 1 and Table 2), with rich data, making it easy for readers to quickly obtain key information. There are a few suggestions for the authors to modify the manuscript.
- The table content is dense, and some data is duplicated (such as multiple studies on the association between DNA methylation and obesity).
- When discussing epigenetic mechanisms, add background knowledge (such as the biological significance of DNA methylation and histone acetylation) are needed, but to avoid excessive terminology.
- And why are histone acetylation and small RNAs related to obesity combined and presented in the table, while methylation is listed separately, as they are all epigenetic factors? Kindly, suggest merging them all into one large summary table or listing them separately.
- Partly, the application of microbiome engineering (such as synthetic biology modified probiotics) and novel metabolites (such as indole derivatives) could be added in the treatment of obesity.
- How about to clearly summarize the core viewpoint (such as' GM-SCFAs epigenetic axis is a key target for intervening in obesity '). Would it be possible to include a figure illustrating the relationship among obesity, the gut microbiome, and epigenetics? Such a visual aid could highlight the central concept of the obesity–epigenetics–microbiome axis proposed in the review and help readers better understand the interactions among these three components.
- The conclusion does not clearly summarize the key findings, and the discussion of future research directions (such as personalized interventions and long-term efficacy verification) is relatively vague.
- Kindly suggest to add footnotes for all abbreviations (such as FFAR=free fatty acid receptor, HDAC=histone deacetylase) in the tables and the charts, because some annotations are unclear (such as "FFAR and HDAC mRNA expression" without clear abbreviations).
- Correct grammar errors (such as ‘alerting histone methylation 'should be' altering histone methylation '). Two different abbreviations for type 2 diabetes appear in lines 19 and 43.
- This review contains a large number of technical terms. It is recommened that including a glossary at the end of the manuscript to help readers quickly understand key concepts during the reading process. Additionally, the abbreviation of a proprietary noun that appears for the first time needs to be completed after writing the full name, to ensure that all technical terms are introduced with their full names at first mention, and that abbreviations are used accurately and consistently throughout the text.
Author Response
Comments and Suggestions for Authors
This study systematically reviewed the interactions between obesity, epigenetics and gut microbiome (GM), covering multidimensional mechanisms such as DNA methylation, histone modification and miRNA regulation. Detailed list of microbiome changes related to obesity (such as Firmicutes/Bacteroidetes imbalance) and treatment strategies (such as probiotics, SCFAs, calorie restriction, exercise). At the same time, the tables are organized clearly (such as Table 1 and Table 2), with rich data, making it easy for readers to quickly obtain key information. There are a few suggestions for the authors to modify the manuscript.
We highly appreciate your feedback on our manuscript. We have revised the manuscript based on your valuable comments, all of which are highlighted in blue for your convenience.
- The table content is dense, and some data is duplicated (such as multiple studies on the association between DNA methylation and obesity).
Author response and action taken: We have addressed the comment in the revised version. We reviewed the tables for redundancy and repetition—please refer to the updated tables. Regarding the multiple studies on the association between DNA methylation and obesity, due to the variety of tissues, animal models, and genes examined, we were unable to include all studies while maintaining clarity and reducing density in each row. However, we have summarized the content in each table to make it more concise
- When discussing epigenetic mechanisms, add background knowledge (such as the biological significance of DNA methylation and histone acetylation) are needed, but to avoid excessive terminology.
Author response and action taken: Relevant information has been added to the manuscript in response to this comment, highlighted in blue under “2. Epigenetics and development of obesity” page 3-4, lines 104-136.
- And why are histone acetylation and small RNAs related to obesity combined and presented in the table, while methylation is listed separately, as they are all epigenetic factors? Kindly, suggest merging them all into one large summary table or listing them separately.
Author response and action taken: We implemented the comment in the revised version and listed tables separately. please see tables 1, 2, and 3.
Table 1. DNA methylation alterations associated with obesity (OB).
Table 2. Histone modifications alterations related to obesity (OB)
Table 3. miRNA alterations related to obesity (OB).
- Partly, the application of microbiome engineering (such as synthetic biology modified probiotics) and novel metabolites (such as indole derivatives) could be added in the treatment of obesity.
Author response and action taken: Relevant information has been added to the manuscript in response to this comment, highlighted in blue under “5.3. 2 Indole and its derivatives for treatment of obesity and obesity-related disorders” page 14, lines 377-393 and “5.5. Engineered Probiotics by Synthetic Biology Approaches for Management of Obesity” page 16-17, lines 467-498.
- How about to clearly summarize the core viewpoint (such as' GM-SCFAs epigenetic axis is a key target for intervening in obesity '). Would it be possible to include a figure illustrating the relationship among obesity, the gut microbiome, and epigenetics? Such a visual aid could highlight the central concept of the obesity–epigenetics–microbiome axisproposed in the review and help readers better understand the interactions among these three components.
Author response and action taken: We have implemented the comment in the revised version and included a figure illustrating the relationship between obesity, the gut microbiome, and epigenetics. Please see figure 1.
- The conclusion does not clearly summarize the key findings, and the discussion of future research directions (such as personalized interventions and long-term efficacy verification) is relatively vague.
Author response and action taken: In the revised version, we have worked to improve the conclusion and the discussion on future research directions. Please see new text highlights in blue in the conclusion section.
- Kindly suggest to add footnotes for all abbreviations (such as FFAR=free fatty acid receptor, HDAC=histone deacetylase) in the tables and the charts, because some annotations are unclear (such as "FFAR and HDAC mRNA expression" without clear abbreviations).
Author response and action taken: We added abbreviation at the end of our manuscript after conclusion. page 20-21, lines 658-695.
- Correct grammar errors (such as ‘alerting histone methylation 'should be' altering histone methylation '). Two different abbreviations for type 2 diabetes appear in lines 19 and 43.
Author response and action taken: Based on your comments, the English language throughout the manuscript has been meticulously revised—including grammar, punctuation, spelling, and rephrasing of unclear sentences—to enhance the overall quality of the manuscript.
- This review contains a large number of technical terms. It is recommenced that including a glossary at the end of the manuscript to help readers quickly understand key concepts during the reading process. Additionally, the abbreviation of a proprietary noun that appears for the first time needs to be completed after writing the full name, to ensure that all technical terms are introduced with their full names at first mention, and that abbreviations are used accurately and consistently throughout the text.
Author response and action taken: In the revised version we added a glossary at the end of the manuscript to help readers quickly understand key concepts during the reading process. Page 21-22, lines 697-734.
Round 2
Reviewer 1 Report
Comments and Suggestions for Authors
The authors have addressed all the comments and suggestions appropriately. The manuscript has been improved in clarity, structure, and scientific content. I am satisfied with the revisions made and I consider the manuscript suitable for publication in its current form.